# *Lactobacillus rhamnosus* Sepsis in a Preterm Infant Following Probiotic Administration: Challenges in Diagnosis

**DOI:** 10.3390/microorganisms13020265

**Published:** 2025-01-25

**Authors:** Ilaria Farella, Maria Fortunato, Domenico Martinelli, Carmela De Carlo, Eleonora Sparapano, Stefania Stolfa, Federica Romanelli, Vittoriana De Laurentiis, Stefano Martinotti, Loredana Capozzi, Stefano Castellana, Antonio Parisi, Giuseppe Latorre

**Affiliations:** 1Department of Medicine and Surgery, LUM University, Casamassima, 70010 Bari, Italy; s.martinotti@miulli.it; 2Neonatology and Neonatal Intensive Care Unit, Ecclesiastical General Hospital F. Miulli, Acquaviva delle Fonti, 70021 Bari, Italy; d.martinelli@miulli.it (D.M.); g.latorre@miulli.it (G.L.); 3Unit of Clinical Pathology and Microbiology, Miulli Regional General Hospital, Acquaviva delle Fonti, 70021 Bari, Italy; m.fortunato@miulli.it; 4UOC Microbiology and Virology, Azienda Ospedaliera-Universitaria Policlinico of Bari, 70124 Bari, Italy; carmela.decarlo@policlinico.ba.it (C.D.C.); eleonora.sparapano@policlinico.ba.it (E.S.); stefania.stolfa@policlinico.ba.it (S.S.); federicarosaromanelli@gmail.com (F.R.); vittoriana.delaurentiis@policlinico.ba.it (V.D.L.); 5Istituto Zooprofilattico Sperimentale della Puglia e della Basilicata, 71121 Foggia, Italy; loredana.capozzi@izspb.it (L.C.); stefano.castellana@izspb.it (S.C.); antonio.parisi@izspb.it (A.P.)

**Keywords:** probiotic sepsis, *Lactobacillus rhamnosus*, preterm infant, whole-genome sequencing, neonatal sepsis

## Abstract

Probiotic administration has become common practice in neonatal intensive care units (NICUs) to prevent necrotizing enterocolitis (NEC) and promote gut health in preterm infants. While probiotics are generally considered safe, rare cases of probiotic-related sepsis have been reported. We present a case of *Lactobacillus rhamnosus* sepsis in a preterm infant, highlighting the challenges involved in its diagnosis. The infant developed symptoms of sepsis on the 13th day of probiotic treatment. Laboratory analyses, including MALDI-TOF, BioFire BCID2 panel, and whole-genome sequencing (WGS), helped confirm the diagnosis and the presence of *Lactobacillus rhamnosus*. In this case, accurately identifying the *Lactobacillus rhamnosus* strain proved challenging, as initial analyses using the Vitek 2 system yielded incorrect identifications. This highlights the limitations of automated systems in distinguishing closely related species, reinforcing the need for advanced molecular techniques to achieve precise strain identification and confirm a probiotic-related infection. Given these diagnostic complexities, it is crucial for clinicians to maintain a high index of suspicion for probiotic-related infections in cases of unexplained sepsis, as this awareness can prompt further diagnostic investigations to ensure accurate pathogen identification. The infant responded to ampicillin therapy, showing clinical improvement within 10 days and was discharged in good health at 67 days of life. This case underscores the importance of advanced molecular diagnostic methods to confirm probiotic-related infections and highlights the need for caution in administering probiotics to vulnerable populations, such as preterm infants. Clinicians must maintain a high index of suspicion for probiotic-associated sepsis in unexplained cases of infection and tailor antibiotic therapy based on susceptibility profiles. These findings emphasize the need for rigorous monitoring, appropriate probiotic strain selection, and optimized safety protocols in NICUs to mitigate potential risks.

## 1. Introduction

*Lactobacillus* species are a group of Gram-positive, rod-shaped facultative anaerobes widely distributed in nature. Known for their role in fermenting carbohydrates into lactic acid, these bacteria are commonly used in the food industry for fermenting dairy products. They also constitute part of the normal commensal flora in the human gastrointestinal tract and have been widely used as probiotics due to their beneficial effects on gut health [1].

Probiotics, including *Lactobacillus rhamnosus*, have been increasingly used in NICUs to prevent necrotizing enterocolitis (NEC) [2]. NEC is a severe gastrointestinal condition that primarily affects preterm and very low birth weight (VLBW) neonates. It is characterized by intestinal inflammation and necrosis, often leading to systemic complications [3]. NEC is the leading cause of gastrointestinal-related mortality in preterm infants with an incidence of 5–12% among VLBW neonates [4]. The diagnosis of NEC is typically guided by Bell’s Modified Staging Criteria, which categorize the condition into stages of escalating severity [5]. Prevention efforts are centered on minimizing risk factors and enhancing intestinal health. Early exposure to colostrum and mother’s milk has demonstrated significant benefits in lowering the incidence of NEC. Additionally, carefully managed feeding protocols with human milk, skin-to-skin care, and the use of probiotics have been identified as effective strategies for promoting gut health and reducing NEC risk [6]. Probiotics, by modulating the intestinal microbiome, have emerged as a particularly promising approach in preventing NEC, as they help establish a healthy microbial balance in the gut, reducing the likelihood of inflammation and associated complications [7]. The results derived from multiple systematic reviews and meta-analyses provide a complex panorama of evidence. Probiotics, particularly multi-strain formulations, are associated with a reduced risk of NEC, late-onset sepsis, and all-cause mortality in preterm infants [2,8]. A recent meta-analysis of RCTs revealed that LGG significantly reduced the risk of NEC stage ≥ II, but it had no significant effect on late-onset sepsis, mortality, time to full feeds, or duration of hospital stay. In contrast, the data from non-RCTs showed no significant effects on NEC, late-onset sepsis, or mortality. The results emphasize that while LGG is effective in reducing the risk of NEC in preterm infants, observational studies did not show the same benefits, highlighting the need for further research to guide clinical practice [9].

However, cases of probiotic-associated sepsis have been reported, particularly in premature neonates. *Lactobacillus rhamnosus* and *Bifidobacterium infantis* are commonly used probiotics, but they have been implicated in cases of bacteremia and sepsis in neonates, raising concerns about their safety in this vulnerable population [10,11].

This case report describes an extremely low birth weight infant who developed *Lactobacillus rhamnosus* sepsis following probiotic supplementation and discusses the complexity of diagnosis and the potential risks of probiotic use in preterm infants.

## 2. Case Description

A white male infant was born at 26 weeks of gestation by vaginal delivery. The infant had a birth weight of 990 g and Apgar scores of 6 and 8 at 1 and 5 min, respectively. He was admitted to the NICU for the management of respiratory distress syndrome and required nasal intermittent positive pressure ventilation. A single dose of surfactant was administered, and he was treated for a patent ductus arteriosus with ibuprofen for 3 days. The infant received daily oral drops of *Lactobacillus rhamnosus* (LGG^®^) (DSM 33156) 1 × 10^9^ CFU, *Bifidobacterium infantis* (DSM 33361) 1 × 10^8^ CFU from the first day of life to prevent NEC. The probiotics were administered through an orogastric tube as part of the unit’s routine protocol. The newborn was fed a combination of breast milk and formula through continuous gastric feeding, which was adjusted proportionally to the weight achieved. On day 13 of life, while on non-invasive respiratory support and with a central venous catheter in place, the infant developed signs of sepsis, including apnea, hypotonia, hyperglycemia, thrombocytopenia (39 × 10^3^/microL), elevated C-reactive protein (3.1 mg/dL) and procalcitonin levels (2.0 ng/mL). After collecting the blood cultures, empiric antibiotic therapy with vancomycin and amikacin was initiated.

### 2.1. Bacterial Isolation

The blood culture taken was incubated in the BACT/ALERT system. After approximately 48 h, the blood culture gave a positive signal. Microscopic evaluation using Gram stain revealed the presence of Gram-positive bacilli. Concurrently, the BioFire^®^ Blood Culture Identification 2 (BCID2) Panel, a multiplex nucleic acid test designed for the BioFire^®^FilmArray^®^2.0 system, was performed. This panel enables the simultaneous detection and identification of various bacteria, yeasts, and genetic determinants associated with antimicrobial resistance commonly involved in bloodstream infections (Table 1).

The BioFire BCID2 panel yielded negative results for all pathogens and resistance genes tested, including methicillin resistance (mecA/C), vancomycin resistance (vanA and vanB), and β-lactam resistance genes (CTX-M, IMP, KPC, NDM, OXA48-LIKE, VIM) (Table 1).

Subcultures were grown using ready-to-use plates with different culture media. For aerobic microorganisms, Wurtz Lactose Agar, Columbia CNA Agar, and Sabouraud Dextrose Agar were incubated at 37 °C for 24–48 h. For anaerobic microorganisms, Columbia Blood Agar, Chocolate Agar, and Schaedler Agar were incubated in the GenBag Anaerobic system at 37 °C for 15 days. After 24 h, aerobic growth was observed on the Columbia CNA Agar plate, and after 48 h, anaerobic growth was detected on Columbia Blood Agar. Microscopic evaluation and Gram staining confirmed the presence of Gram-positive bacilli. Using the Vitek 2 system, initial identification suggested *Pediococcus acidilactici* with 86% probability, but this was deemed inconsistent with the microscopic findings. Further anaerobic growth was observed, and the isolate was later identified as *Lactobacillus* with an 87% probability using the VITEK 2 ANC ID card. However, due to the limitations of the VITEK 2 system in distinguishing between closely related species, the positive blood culture was sent to the Microbiology Unit of the Bari Polyclinic for further gene-sequencing analysis. The umbilical venous catheter was cultured and tested negative.

### 2.2. 16S rDNA Gene Sequencing

To confirm the species identification further, 16S rDNA gene sequencing was performed on both the clinical and probiotic isolates. The obtained sequences were analyzed using NCBI BLAST (https://blast.ncbi.nlm.nih.gov/Blast.cgi, accessed on 25 September 2023), yielding a 99.46% identity with *Lactobacillus rhamnosus*.

### 2.3. Whole-Genome Sequencing and Bioinformatic Analyses

Whole-genome sequencing (WGS) was performed to compare the clinical isolate from the patient’s blood with the probiotic strain. The WGS data revealed two shared mutations between the isolates, which were located in the PTS glucose transporter gene and an intergenic region, strongly suggesting that the strain isolated from the blood culture was identical to the probiotic strain administered to the patient. This analysis identified a mutation in the PTS glucose transporter subunit IIABC gene, resulting in a missense mutation (L422P), which could potentially alter glucose transport within the bacterium. Additionally, another mutation was found in an intergenic region, which may influence gene regulation and could have broader implications for the organism’s function. These findings provide strong genetic evidence linking the administered probiotic strain to the bloodstream infection in the patient (Table 2).

### 2.4. Antibiotic Susceptibility Testing

Antibiotic susceptibility testing was conducted using the E-test method. The *Lactobacillus rhamnosus* isolate showed susceptibility to penicillin, ampicillin, and clindamycin but resistance to meropenem and metronidazole. These results were interpreted based on the European Committee on Antimicrobial Susceptibility Testing (EUCAST) guidelines (Table 3).

After the blood culture tested positive, probiotic supplementation was discontinued, and antibiotics were switched to ampicillin, which was administered for 10 days. The patient recovered gradually and was extubated 2 days after starting ampicillin therapy. The inflammatory markers normalized after 72 h of antibiotic therapy. Platelet count normalized within 10 days, and his clinical condition progressively improved. Ultimately, the patient was discharged in good health at 67 days of life.

### 2.5. Timeline of Clinical Interventions and Major Diagnostic Steps

−Day 1: Start probiotic (LGG + B. infantis); partial enteral feeding initiated.−Day 3: Ibuprofen for PDA closure completed.−Day 13: Clinical signs of sepsis; blood cultures drawn; vancomycin + amikacin started.−Day 13–14: Blood culture yields Gram-positive rods; automated system (Vitek 2) misidentifies the strain.−Day 15: Advanced molecular tests confirm *Lactobacillus rhamnosus*; probiotic discontinued.−Day 15: Antibiotic therapy switched to ampicillin for 10 days.−Day 17: Clinical improvement.−Day 27: Normalization of platelet count.

## 3. Discussion

*Lactobacillus* species are Gram-positive, non-spore-forming, rod-shaped facultative anaerobes that play a key role in the fermentation of sugars into lactic acid. These bacteria are widely recognized for their beneficial probiotic effects on gut health and are extensively used in the food industry, particularly in the production of fermented foods. Many strains of *Lactobacillus* are part of the commensal flora of the human gut and genital tract, providing beneficial effects such as preventing inflammation and infection. These bacteria are highly diverse, and their identification often requires molecular methods due to their taxonomical complexity [12].

The use of probiotics, including *Lactobacillus* species, in neonatal care, particularly for preterm infants, has gained popularity due to their ability to modulate the intestinal microbiome and reduce the incidence of conditions like NEC and late-onset sepsis [10]. ESPGHAN has provided recommendations regarding which strains to use, the optimal dosage, and treatment duration, highlighting the importance of specific strains such as *Lactobacillus rhamnosus* GG ATCC53103 or the combination of *Bifidobacterium infantis* Bb-02, *Bifidobacterium lactis* Bb-12, and *Streptococcus thermophilus* TH-4 to reduce NEC rates [13]. Systematic reviews have shown that multi-strain probiotic products or combinations with synbiotics are more effective in preventing morbidity and mortality compared to single-strain probiotics [14]. However, the use of probiotics is not without risk especially in vulnerable populations such as preterm infants. In the United States, the FDA has raised concerns about the manufacture and regulation of these products, often classifying them alongside dietary supplements rather than subjecting them to the same approval process as pharmaceuticals. Consequently, the American Academy of Pediatrics has taken a more cautious stance, limiting the widespread use of probiotics in preterm neonates.

Although available data indicate that probiotics can lower the incidence of NEC and mortality, the precise rate of probiotic-related sepsis remains unclear [15,16]. Therefore, it has been proposed that the decision to administer probiotics to preterm and/or very low birth weight infants should be an informed, shared process involving parents. This “shared decision-making” model ensures families understand the complexity of therapeutic options in alignment with their values and preferences. Meanwhile, ongoing research and clinical trials (including those testing pharmaceutical-grade products) will help clarify the risk–benefit profile of probiotics and may drive stricter regulatory requirements in the future [17].

In our case, we report a preterm infant who developed *Lactobacillus rhamnosus GG* sepsis 13 days after starting probiotic supplementation with a product containing *Lactobacillus rhamnosus* and *Bifidobacterium infantis*. The onset of sepsis was characterized by symptoms including apnea, hypotonia, thrombocytopenia, and elevated inflammatory markers, requiring empirical antibiotic therapy. Blood cultures revealed the presence of *Lactobacillus rhamnosus*, and subsequent whole-genome sequencing confirmed the identity of the strain as identical to the probiotic strain administered. These findings support the cause–effect relationship between the probiotic administration and the development of sepsis, highlighting the potential risk in using probiotics in critically ill neonates.

A crucial aspect to consider is the difficulty in accurately identifying the *Lactobacillus rhamnosus* bacterial strain, which requires targeted clinical suspicion to avoid incomplete diagnosis. In this case, initial analyses conducted with the Vitek 2 system suggested an erroneous identification of *Pediococcus acidilactici* (with an 86% probability) and subsequently of *Lactobacillus casei* (87% probability). The limitation of automated systems like Vitek 2 in distinguishing between closely related species necessitates molecular techniques such as whole-genome sequencing and 16S rDNA gene sequencing for accurate strain identification and to establish the link between the probiotic and the infection. This need highlights the importance of alerting the microbiology laboratory when probiotics are administered to high-risk patients so that appropriate identification methods can be used [18].

Probiotic-associated bacteremia in neonates is rare but has been documented in several case reports [19].

The exact mechanism of probiotic sepsis is not fully understood, but several primary routes have been suggested, including intestinal translocation, blood contamination during probiotic preparation and administration, and catheter-related infections. The latter mechanism appears more common due to *Lactobacillus rhamnosus GG*’s capacity to form biofilms. A review of *Lactobacillus rhamnosus GG* sepsis in preterm and young infants revealed that nearly 60% of cases involved a suspected or confirmed infected central venous catheter. In these instances, bacterial translocation across the immature gut barrier likely results in hematogenous catheter contamination with the administration of glucose through the catheter creating a favorable environment for biofilm formation and subsequent bloodstream infection [10].

Premature infants, due to their immature immune systems and underdeveloped gut barriers, are particularly vulnerable to infections. In this case, bacterial dissemination likely occurred through hematogenous spread, facilitated by an overly permeable gut, which is a condition referred to as “leaky gut” in preterm infants. Previous studies have demonstrated that intestinal permeability is higher in preterm infants compared to term infants with no significant differences based on gestational age before 34 weeks of completed gestation [20,21]. Intestinal permeability tends to decrease with the initiation of enteral feeding. Notably, intestinal permeability in preterm infants is significantly reduced for those receiving human milk compared to formula in a dose-related manner during the first postnatal month. Furthermore, the fecal proteome of preterm infants is deprived of gastrointestinal barrier-related proteins during the first six postnatal weeks compared to term infants [22]. These findings suggest that the immaturity of the preterm gut, combined with feeding type and other factors, may contribute to increased vulnerability to infections and bacterial spread through a permeable gut barrier [23].

Previous studies have shown that *Lactobacillus* species, while generally safe, can act as opportunistic pathogens in vulnerable populations, leading to infections such as bacteremia, endocarditis, and abscesses [12]. In particular, *Lactobacillus rhamnosus GG* has been implicated in cases of sepsis and other systemic infections in neonates, which are often associated with the use of probiotic supplements [18]. A systematic review of 1569 studies found only 32 cases of *Lactobacillus* or *Bifidobacterium* sepsis in preterm neonates, which demonstrates the rarity of this complication but also underlines the need for careful monitoring when administering probiotics to high-risk patients [24].

Antibiotic susceptibility testing in this case revealed that the *Lactobacillus rhamnosus* strain was sensitive to penicillin and ampicillin but resistant to vancomycin and meropenem. These findings are consistent with previous reports of antibiotic susceptibility in *Lactobacillus* species, which are typically resistant to glycopeptides like vancomycin. This poses a challenge for empirical antibiotic therapy in neonatal units where vancomycin is often used as a first-line treatment for suspected sepsis. Our findings emphasize the need for clinicians to consider alternative antibiotic options, such as penicillins or aminoglycosides, in cases of suspected probiotic-related sepsis [10].

Furthermore, the difficulty in identifying probiotic strains like *Lactobacillus rhamnosus* using automated culture systems is an important consideration. In this case, traditional identification methods such as the Vitek 2 system failed to accurately identify the probiotic strain, necessitating the use of molecular techniques such as whole-genome sequencing and 16S rDNA sequencing. As reported by Bosshard et al., 16S rRNA gene sequencing was more accurate, assigning 92% of isolates at the species level compared to 54% using Vitek 2 [25]. In our case, 16S rDNA sequencing was crucial in confirming the identity of the strain and establishing the link between the probiotic and the infection. This highlights the importance of alerting the microbiology laboratory when probiotics are used, especially in high-risk patients, to ensure that appropriate identification methods are employed [18]. In clinical settings, a practical solution is to alert the microbiology laboratory when a patient is receiving probiotics so that alternative or confirmatory methods (e.g., MALDI-TOF, 16S rRNA sequencing, or WGS) can be employed promptly. Additionally, Gram staining could be utilized, as it may aid in the identification of the bacterium. This multimodal diagnostic approach helps ensure correct species-level identification and allows clinicians to tailor antibiotic therapy accurately. In addition, it could be interesting to add to traditional inflammatory markers, such as PCR and procalcitonin, advanced techniques like RNA transcriptomic signatures, which could represent a significant advancement in diagnosing infections in preterm neonates. These innovative tools, although still not extensively studied in neonates, could help reduce inappropriate antibiotic use and its associated risks, such as antibiotic resistance and negative impacts on the neonatal microbiota. [26].

Despite the proven benefits of probiotics in reducing NEC, this case underscores the need for more stringent guidelines on probiotic administration and monitoring in preterm infants, particularly those with central lines and immature gut barriers. The risk of sepsis, though low, is not negligible, and a clearer framework on strain selection, dosing, and duration of therapy may help minimize complications while preserving the established benefits of probiotic use.

## 4. Conclusions

Although probiotics have demonstrated significant benefits in preventing NEC and other complications in preterm infants, their use is not without risks. Rare but documented cases of probiotic-associated sepsis, particularly in neonates, underscore the importance of cautious administration and vigilant monitoring in this vulnerable population. Laboratory support is critical to accurately identify infections associated with probiotics, as misleading results from automated systems can delay appropriate therapy. To address this challenge, clinicians and microbiologists should ensure that laboratories are informed when a patient is receiving probiotic supplementation. Molecular methods, such as 16S rRNA sequencing or WGS, should be considered when *Lactobacillus* sepsis is suspected to ensure precise pathogen identification. The management of probiotic-associated sepsis also requires antibiotic regimens tailored to the specific susceptibility profiles of the implicated strains, especially given the potential for resistance to commonly used antibiotics like vancomycin. Further research is essential to establish standardized safety protocols for probiotic use in neonatal intensive care units. These protocols should include recommendations on strain selection, optimal dosing, and the duration of administration alongside strategies to minimize contamination risks during preparation and delivery. Close monitoring and individualized patient assessment remain pivotal to ensuring both the safety and efficacy of probiotic use in critically ill or extremely preterm infants. Moreover, considering probiotic-associated sepsis in the differential diagnosis of sepsis is crucial for timely and appropriate treatment, ultimately improving outcomes for this fragile patient group.

## Figures and Tables

**Table 1 microorganisms-13-00265-t001:** Pathogens detected by BioFire^®^ Blood Culture Identification 2 (BCID2) Panel.

Yeasts	Gram-Negative Bacteria	Gram-Positive Bacteria
*Candida albicans*	*Acinetobacter baumannii*	*Enterococcus faecalis*
*Candida auris*	*Bacteroides fragilis*	*Enterococcus faecium*
*Candida glabrata*	*Haemophilus influenzae*	*Listeria monocytogenes*
*Candida krusei*	*Neisseria meningitidis*	*Staphylococcus* spp.
*Candida parapsilosis*	*Pseudomonas aeruginosa,*	*Staphylococcus aureus*
*Candida tropicalis*	*Stenotrophomonas maltophilia*	*Staphylococcus epidermidis*
*Cryptococcus neoformans*/*gattii*	*Enterobacterales*	*Staphylococcus lugdunensis*
	*Enterobacter cloacae*	*Streptococcus* spp.
	*Escherichia coli*	*Streptococcus agalactiae*
	*Klebsiella aerogenes*	*Streptococcus pneumoniae*
	*Klebsiella oxytoca*	*Streptococcus pyogenes*
	*Klebsiella pneumoniae*	
	*Proteus* spp.	
	*Salmonella* spp.	
	*Serratia marcescens*	

**Table 2 microorganisms-13-00265-t002:** Genetic variants identified in *Lactobacillus rhamnosus* from blood culture and probiotic supplement.

Chr Position	Ref nt	Alt nt	Gene/Effect
615,483	T	C	PTS glucose transporter subunit IIABC (L422P)
1,883,242	C	A	Intergenic region

**Legend:** Alt nt, alternate nucleotide; Chr, chromosomal; Gene/Effect, gene affected and the nature of the mutation; Ref nt, reference nucleotide.

**Table 3 microorganisms-13-00265-t003:** Antibiotic susceptibility testing using the E-test method.

Antibiotics	Blood Culture Isolate MIC (µg/mL)	Probiotic Isolate MIC (µg/mL)
Benzylpenicillin	0.50	0.50
Piperacillin/Tazobactam	1	1
Ampicillin/Sulbactam	1.5	1.5
Clindamycin	0.19	0.19
Meropenem	8	8
Metronidazole	>256	>256
Linezolid	0.094	2

## Data Availability

The original contributions presented in this study are included in the article. Further inquiries can be directed to the corresponding author.

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
