# Peer review of "Lactobacillus rhamnosus Sepsis in a Preterm Infant Following Probiotic Administration: Challenges in Diagnosis"

_microorganisms, 2025, doi:10.3390/microorganisms13020265_

Round 1
Reviewer 1 Report
Comments and Suggestions for Authors
I have read this case report with great interest. While the case report in its clinical part is indeed confirming previous case reports or series on lactobacillus rhamnosus sepsis, the experience reported on the laboratory limitations of automatic analyses brings additional value to the current paper.
Is the suppletion standard practice in your unit ?
Just as a suggestion, there is perhaps value to provide an overview on the cases until currently reported, although I understand that this distracts from the key messages of your paper.
Author Response
REVIEWER 1
I would like to express my sincere gratitude for your valuable suggestions and corrections.
I HAVE READ THIS CASE REPORT WITH GREAT INTEREST. WHILE THE CASE REPORT IN ITS CLINICAL PART IS INDEED CONFIRMING PREVIOUS CASE REPORTS OR SERIES ON LACTOBACILLUS RHAMNOSUS SEPSIS, THE EXPERIENCE REPORTED ON THE LABORATORY LIMITATIONS OF AUTOMATIC ANALYSES BRINGS ADDITIONAL VALUE TO THE CURRENT PAPER.
IS THE SUPPLETION STANDARD PRACTICE IN YOUR UNIT ?
Thank you very much for your interest in our clinical practice. Before the experience described in the case report, it was our usual practice to supplement all preterm infants with probiotics. Currently, the administration of probiotics is discussed on a case-by-case basis.
JUST AS A SUGGESTION, THERE IS PERHAPS VALUE TO PROVIDE AN OVERVIEW ON THE CASES UNTIL CURRENTLY REPORTED, ALTHOUGH I UNDERSTAND THAT THIS DISTRACTS FROM THE KEY MESSAGES OF YOUR PAPER.
I greatly appreciate your suggestion to include an overview of recent cases. I have found a highly relevant study that addresses your request, which I have incorporated into the discussion section. The study, titled “Lactobacillus rhamnosus GG as a probiotic for preterm infants: a strain-specific systematic review and meta-analysis; DOI: 10.1038/s41430-024-01474-0
Reviewer 2 Report
Comments and Suggestions for Authors
Farella et al submitted an interesting case report discussing Lactobacillus rhamnosus sepsis in a preterm infant following probiotic administration, highlighting the challenges of accurate diagnosis and the risks associated with probiotic use in vulnerable populations. The study addresses the rare but critical issue of probiotic-associated sepsis in preterm infants, focusing on The diagnostic challenges of distinguishing probiotic strains from pathogenic isolates and the clinical implications of using probiotics in immunocompromised populations. The manuscript presents a clinically significant topic with well-documented findings; however, I would like to add some minor suggestions/comments to enhance the clarity and overall impact of the manuscript prior to the publication.
1. The abstract should include quantified results, such as the exact timeline and diagnostic findings, to make it more precise and informative. Also, it’d better that authors consider explicitly stating the implications for clinical practice to attract a broader readership.
2- The introduction lacks a critical evaluation of existing studies on probiotic-associated sepsis. Here is my suggestion: authors should include more recent meta-analyses or systematic reviews to strengthen the background. Also, authors are better to highlight the gap this case addresses more explicitly, such as challenges in strain identification and its clinical implications.
3- In the case description, the discussion on laboratory findings could be more concise, focusing on the key diagnostic challenges. I would also suggest that authors include a clearer timeline of interventions, diagnostic milestones, and clinical improvements for better readability.
4- The discussion should have more details on the limitations of automated systems (e.g., Vitek 2) versus advanced molecular techniques as well as suggested practical solutions for clinical settings. Authors are better to compare this case with similar published cases, emphasizing what makes it unique and then discuss the potential impact of this case on neonatal care guidelines, particularly regarding the use of probiotics in high-risk groups.
5- the conclusion could be expanded to include specific recommendations for clinicians and microbiologists regarding probiotic administration and monitoring.
Author Response
I would like to express my sincere gratitude for your valuable suggestions and corrections.
- THE ABSTRACT SHOULD INCLUDE QUANTIFIED RESULTS, SUCH AS THE EXACT TIMELINE AND DIAGNOSTIC FINDINGS, TO MAKE IT MORE PRECISE AND INFORMATIVE. ALSO, IT’D BETTER THAT AUTHORS CONSIDER EXPLICITLY STATING THE IMPLICATIONS FOR CLINICAL PRACTICE TO ATTRACT A BROADER READERSHIP.
I have revised the abstract as per your suggestion and incorporated the requested changes: “The infant responded to ampicillin therapy, showing clinical improvement within 10 days and was discharged in good health at 67 days of life”. This case highlights the need for careful monitoring of probiotic use in vulnerable populations like preterm infants, as well as the importance of precise bacterial identification methods to ensure accurate diagnosis and treatment. “This case underscores the importance of advanced molecular diagnostic methods to confirm probiotic-related infections and highlights the need for caution in administering probiotics to vulnerable populations, such as preterm infants. Clinicians must maintain a high index of suspicion for probiotic-associated sepsis in unexplained cases of infection and tailor antibiotic therapy based on susceptibility profiles. These findings emphasize the need for rigorous monitoring, appropriate probiotic strain selection, and optimized safety protocols in NICUs to mitigate potential risks.”
2- THE INTRODUCTION LACKS A CRITICAL EVALUATION OF EXISTING STUDIES ON PROBIOTIC-ASSOCIATED SEPSIS. HERE IS MY SUGGESTION: AUTHORS SHOULD INCLUDE MORE RECENT META-ANALYSES OR SYSTEMATIC REVIEWS TO STRENGTHEN THE BACKGROUND. ALSO, AUTHORS ARE BETTER TO HIGHLIGHT THE GAP THIS CASE ADDRESSES MORE EXPLICITLY, SUCH AS CHALLENGES IN STRAIN IDENTIFICATION AND ITS CLINICAL IMPLICATIONS.
In the introduction we add : “A recent meta-analysis of RCTs revealed that LGG significantly reduced the risk of NEC stage ≥ II, but had no significant effect on late-onset sepsis, mortality, time to full feeds, or duration of hospital stay. In contrast, the data from non-RCTs showed no significant effects on NEC, LOS, or mortality. The results emphasize that while LGG is effective in reducing the risk of NEC in preterm infants, observational studies did not show the same benefits, highlighting the need for further research to guide clinical practice (Ananthan, 2024).
3- IN THE CASE DESCRIPTION, THE DISCUSSION ON LABORATORY FINDINGS COULD BE MORE CONCISE, FOCUSING ON THE KEY DIAGNOSTIC CHALLENGES. I WOULD ALSO SUGGEST THAT AUTHORS INCLUDE A CLEARER TIMELINE OF INTERVENTIONS, DIAGNOSTIC MILESTONES, AND CLINICAL IMPROVEMENTS FOR BETTER READABILITY.
Thank you for highlighting the excessive verbosity in the description of the laboratory data. We have streamlined the section related to laboratory analyses and added a brief chronological overview (timeline) of the clinical interventions and the main diagnostic and therapeutic events.
4- THE DISCUSSION SHOULD HAVE MORE DETAILS ON THE LIMITATIONS OF AUTOMATED SYSTEMS (E.G., VITEK 2) VERSUS ADVANCED MOLECULAR TECHNIQUES AS WELL AS SUGGESTED PRACTICAL SOLUTIONS FOR CLINICAL SETTINGS. AUTHORS ARE BETTER TO COMPARE THIS CASE WITH SIMILAR PUBLISHED CASES, EMPHASIZING WHAT MAKES IT UNIQUE AND THEN DISCUSS THE POTENTIAL IMPACT OF THIS CASE ON NEONATAL CARE GUIDELINES, PARTICULARLY REGARDING THE USE OF PROBIOTICS IN HIGH-RISK GROUPS.
We have expanded the discussion: “As reported by Bosshard et al. 16S rRNA gene sequencing was more accurate, assigning 92% of isolates at the species level, compared to 54% Vitek 2 {Bosshard, 2006}.”; “In clinical settings, a practical solution is to alert the microbiology laboratory when a patient is receiving probiotics so that alternative or confirmatory methods (e.g., MALDI-TOF, 16S rRNA sequencing, or WGS) can be employed promptly. Additionally, Gram staining could be utilized, as it may aid in the identification of the bacterium. This multimodal diagnostic approach helps ensure correct species-level identification and allows clinicians to tailor antibiotic therapy accurately.” ; “However, the use of probiotics is not without risk, especially in vulnerable populations such as preterm infants. In the United States, the FDA has raised concerns about the manufacture and regulation of these products, often classifying them alongside dietary supplements rather than subjecting them to the same approval process as pharmaceuticals. Consequently, the American Academy of Pediatrics has taken a more cautious stance, limiting the widespread use of probiotics in preterm neonates.
This discrepancy underscores different perspectives on weighing the benefits (such as reduced NEC or mortality) against potential risks (including sepsis, contamination, and production uncertainties). Although available data indicate that probiotics can lower the incidence of NEC and mortality, the precise rate of probiotic-related sepsis remains unclear {van den Akker, 2018 } {Sharif, 2023 }. Therefore, it has been proposed that the decision to administer probiotics to preterm and/or very low birth weight infants should be an informed, shared process involving parents. This “shared decision-making” model ensures families understand the complexity of therapeutic options in alignment with their values and preferences. Meanwhile, ongoing research and clinical trials (including those testing pharmaceutical-grade products) will help clarify the risk–benefit profile of probiotics and may drive stricter regulatory requirements in the future {Preidis, 2025 #746}.”
5- THE CONCLUSION COULD BE EXPANDED TO INCLUDE SPECIFIC RECOMMENDATIONS FOR CLINICIANS AND MICROBIOLOGISTS REGARDING PROBIOTIC ADMINISTRATION AND MONITORING.
We have expanded the conclusion by highlighting the need for standardized protocols for probiotic use.
Reviewer 3 Report
Comments and Suggestions for Authors
The authors reported an intriguing case of Lactobacillus rhamnosus sepsis in a preterm infant, a rare but potentially life-threatening infection in neonatal intensive care unit (NICU) patients, particularly in those who are small for gestational age (SGA). Below are my comments:
Line 47 – Please specify the acronym "NEC."
Line 49 – Regarding NEC, it would be beneficial for readers to include a brief introduction to the condition (diagnostic criteria, prevention strategies, etc.).
Line 63 – Should this be AGA or SGA?
Line 74 – If possible, provide more details about the infant's laboratory findings. How elevated was the C-reactive protein (CRP)? Were procalcitonin levels indicative of sepsis or borderline? Additionally, include more information on the clinical course in Lines 137–139, with specific details on the progression of inflammatory markers.
Line 166 – Based on the case report described, in the NEC prevention algorithm, would it be advisable to avoid probiotic supplementation in certain patient categories, such as preterm infants with suspected immunodeficiency?
Line 243 – In cases of suspected probiotic-related sepsis, should the first-line antibiotic therapy follow the standard protocols for late-onset or early-onset sepsis, or is a specific protocol required?
Regarding host-RNA transcript signatures, in recent years it has become evident that they provide significant diagnostic benefits in pediatric infections (10.1038/s41390-021-01890-z). Could they be useful in the early diagnosis of probiotic-associated infections?
Minor improvements to the English translation are necessary.
Author Response
I would like to express my sincere gratitude for your valuable suggestions and corrections.
The authors reported an intriguing case of Lactobacillus rhamnosus sepsis in a preterm infant, a rare but potentially life-threatening infection in neonatal intensive care unit (NICU) patients, particularly in those who are small for gestational age (SGA). Below are my comments:
LINE 47 – PLEASE SPECIFY THE ACRONYM "NEC."
Done
LINE 49 – REGARDING NEC, IT WOULD BE BENEFICIAL FOR READERS TO INCLUDE A BRIEF INTRODUCTION TO THE CONDITION (DIAGNOSTIC CRITERIA, PREVENTION STRATEGIES, ETC.).
We add “NEC is a severe gastrointestinal condition that primarily affects preterm and very low birth weight (VLBW) neonates. It is characterized by intestinal inflammation and necrosis, often leading to systemic complications. NEC is the leading cause of gastrointestinal-related mortality in preterm infants, with an incidence of 5-12% among VLBW neonates. Diagnosis of NEC is typically guided by Bell’s Modified Staging Criteria, which categorize the condition into stages of escalating severity. Prevention efforts are centered on minimizing risk factors and enhancing intestinal health. Early exposure to colostrum and mother’s milk has demonstrated significant benefits in lowering the incidence of NEC. Additionally, carefully managed feeding protocols with human milk, skin-to-skin care, and the use of probiotics have been identified as effective strategies for promoting gut health and reducing NEC risk. Probiotics, by modulating the intestinal microbiome, have emerged as a particularly promising approach in preventing NEC, as they help establish a healthy microbial balance in the gut, reducing the likelihood of inflammation and associated complications”
LINE 63 – SHOULD THIS BE AGA OR SGA?
We add AGA in the text
LINE 74 – IF POSSIBLE, PROVIDE MORE DETAILS ABOUT THE INFANT'S LABORATORY FINDINGS. HOW ELEVATED WAS THE C-REACTIVE PROTEIN (CRP)? WERE PROCALCITONIN LEVELS INDICATIVE OF SEPSIS OR BORDERLINE? ADDITIONALLY, INCLUDE MORE INFORMATION ON THE CLINICAL COURSE IN LINES 137–139, WITH SPECIFIC DETAILS ON THE PROGRESSION OF INFLAMMATORY MARKERS.
We add: line 74- elevated C-reactive protein (3.1 mg/dl) and procalcitonin levels (2.0 ng/ml). ; lines 137-139 The inflammatory markers normalized after 72 hours of antibiotic therapy.
LINE 166 – BASED ON THE CASE REPORT DESCRIBED, IN THE NEC PREVENTION ALGORITHM, WOULD IT BE ADVISABLE TO AVOID PROBIOTIC SUPPLEMENTATION IN CERTAIN PATIENT CATEGORIES, SUCH AS PRETERM INFANTS WITH SUSPECTED IMMUNODEFICIENCY?
At present, no generally adopted standards exist regarding the application of probiotics in patients diagnosed with organ dysfunction, immunodeficiency, and intestinal barrier dysfunction (doi: 10.1002/14651858.CD005496.pub6.); (doi: 10.1111/j.1469-0691.2011.03614.x)
LINE 243 – IN CASES OF SUSPECTED PROBIOTIC-RELATED SEPSIS, SHOULD THE FIRST-LINE ANTIBIOTIC THERAPY FOLLOW THE STANDARD PROTOCOLS FOR LATE-ONSET OR EARLY-ONSET SEPSIS, OR IS A SPECIFIC PROTOCOL REQUIRED?
Recent studies explore various aspects of probiotic use in neonatal care but do not directly address whether the first-line antibiotic therapy for suspected probiotic-related sepsis should adhere to standard protocols for late-onset or early-onset sepsis, or if a specific protocol is necessary. However, some general insights can be inferred. In neonatal sepsis, standard practice typically involves initiating broad-spectrum antibiotics to cover the most common pathogens, and this approach would likely be appropriate initially for suspected probiotic-related sepsis. If, however, there is suspicion that a probiotic strain is responsible for the infection, identifying the organism through blood culture and conducting susceptibility testing would become essential. Such tests would guide the adjustment of antibiotic therapy to target the specific organism effectively, especially if resistance patterns differ from typical pathogens.
REGARDING HOST-RNA TRANSCRIPT SIGNATURES, IN RECENT YEARS IT HAS BECOME EVIDENT THAT THEY PROVIDE SIGNIFICANT DIAGNOSTIC BENEFITS IN PEDIATRIC INFECTIONS (10.1038/S41390-021-01890-Z). COULD THEY BE USEFUL IN THE EARLY DIAGNOSIS OF PROBIOTIC-ASSOCIATED INFECTIONS?
We add: 276-281 In addition it could be interested to add to traditional inflammatory markers, such as PCR and procalcitonin, advanced techniques like RNA transcriptomic signatures could represent a significant advancement in diagnosing infections in preterm neonates. These innovative tools, although still not extensively studied in neonates, could help reduce inappropriate antibiotic use and its associated risks, such as antibiotic resistance and negative impacts on the neonatal microbiota. {Buonsenso, 2022 #742}.
Round 2
Reviewer 3 Report
Comments and Suggestions for Authors
I believe the manuscript can be accepted in its current form, pending the Editor's final decision. I thank the authors for addressing all my comments, which have been adequately addressed.